# Uncovering Resistance to Hepatitis C Virus Infection: Scientific Contributions and Unanswered Questions in the Irish Anti-D Cohort

**DOI:** 10.3390/pathogens11030306

**Published:** 2022-02-28

**Authors:** Jamie A. Sugrue, Cliona O’Farrelly

**Affiliations:** 1School of Biochemistry and Immunology, Trinity Biomedical Sciences Institute, Trinity College Dublin, D02R590 Dublin, Ireland; 2School of Medicine, Trinity College Dublin, D02R590 Dublin, Ireland

**Keywords:** Hepatitis C virus, viral resistance, inter individual variation, exposed seronegative, anti-D cohort

## Abstract

Infections caused inadvertently during clinical intervention provide valuable insight into the spectrum of human responses to viruses. Delivery of hepatitis C virus (HCV)-contaminated blood products in the 70s (before HCV was identified) have dramatically increased our understanding of the natural history of HCV infection and the role that host immunity plays in the outcome to viral infection. In Ireland, HCV-contaminated anti-D immunoglobulin (Ig) preparations were administered to approximately 1700 pregnant Irish rhesus-negative women in 1977–1979. Though tragic in nature, this outbreak (alongside a smaller episode in 1993) has provided unique insight into the host factors that influence outcomes after HCV exposure and the subsequent development of disease in an otherwise healthy female population. Despite exposure to highly infectious batches of anti-D, almost 600 of the HCV-exposed women have never shown any evidence of infection (remaining negative for both viral RNA and anti-HCV antibodies). Detailed analysis of these individuals may shed light on innate immune pathways that effectively block HCV infection and potentially inform us more generally about the mechanisms that contribute to viral resistance in human populations.

## 1. Introduction

Hepatitis C virus (HCV) is a human hepatotropic pathogen, transmitted via direct contact with infectious blood or blood-derived products [1]. Viral transmission that occurs during the course of medical examination or treatment, referred to as iatrogenic infection, was a significant contributor to the historical spread of HCV infection worldwide during the 1970s, 1980s and 1990s, prior to the identification of HCV in 1989 [2,3]. The lack of specific molecular or immune-based detection assays for HCV resulted in several instances of transmission due to the medical use of contaminated blood or blood-derived products, most frequently in haemophiliac populations, and iatrogenic transmission remains a risk in both developing and developed countries [4,5]. The introduction of effective screening for HCV in the 1990s resulted in the disappearance of iatrogenic HCV infections across the globe. Those at high risk of infection with HCV nowadays are intravenous drug users who exchange needles [6].

With the development of diagnostic assays specific for HCV, several countries initiated large scale studies to identify and test individuals exposed to infectious viruses, such as HCV via contaminated blood donations or blood-derived products [7,8,9]. These studies attempted to trace the recipients of HCV-positive blood donations and blood-derived products, screening all individuals who were potentially exposed to these products. In doing so, these studies identified a spectrum of outcomes following HCV exposure that ranged from those who resisted infection (exposed seronegative; ESN), spontaneous resolvers (SR) and those who became chronically infected (CI; Figure 1) [8].

In Ireland, a major iatrogenic transmission event occurred between 1977–1979 (a smaller event occurred in 1991–1994 and is not the focus of this review). The outbreak centred on the distribution of HCV-contaminated anti-D immunoglobulin (Ig) preparations to pregnant Irish rhesus-negative women [10,11,12]. Prophylactic anti-D Ig is administered to mitigate the risk of haemolytic disease of the foetus and new-born infant, which can lead to foetal anaemia, jaundice, and stillbirth [13]. The Irish anti-D Ig HCV outbreak occurred following the use of HCV-infected plasma from a single donor who had symptoms or risk factors indicative of potential HCV infection [12]. This review explores the unique circumstances surrounding the 1977–1979 outbreak and the subsequent research undertaken in these exposed rhesus-negative women. We also point towards novel avenues of exploration to be undertaken in these cohorts, namely an understanding of resistance to HCV infection.

## 2. Uniqueness of the Irish Anti-D Cohort

Animal studies of viral infection require the attainment of a defined amount of a defined infectious agent at the same time to all study participants. Studies that emulate this set up in humans (challenge trials), such as these, are contentious and rare [14]. However, the Irish anti-D-related HCV outbreak replicates the key criteria of the ideal infection study. These key aspects provided a silver-lining to this national tragedy, enabling research that has fundamentally changed how we view and investigate HCV infection worldwide.

The outbreak between the years 1977–1979 was linked to a single HCV-infected individual, whose blood donation was used in the preparation of anti-D Ig [12]. Viruses of the single, contaminating HCV strain were distributed amongst hundreds of batches of anti-D Ig, and administered to thousands of Irish women (Table 1). All the exposed individuals were women of child-bearing age (median age of 28) and were predominantly of an Irish ethnic origin. Collectively, this group of individuals were in relatively good health, with low rates of alcohol consumption, and few lifestyle risk factors associated with HCV transmission [15]. These factors resulted in a highly homogeneous cohort of individuals with similar exposure histories and lengths of infection.

A total of 4062 batches of potentially contaminated anti-D Ig was produced during the 1977–1979 outbreaks, [11,16]. There was also significant variability in infectivity amongst different anti-D Ig batches associated with the 1977–1979 outbreak; six batches in particular were highly infectious [16]. Amongst those individuals that became chronically infected, it has been estimated that 13.9% had signs of liver disease, 11.3% had liver cirrhosis, 1.2% developed liver cancer or hepatocellular carcinoma (HCC), and 2.6% died from liver disease [17].

## 3. What Has Been Learned from the Irish HCV Anti-D Cohort?

Since the widespread recognition of the outbreak in the early 1990s, a significant body of research has been undertaken. This includes almost 70 primary peer-reviewed research articles published in leading international academic and clinical journals, including Gut, NEJM, The Lancet, Hepatology, PNAS, and Gastroenterology. These studies contributed to an epidemiological evidence-base that has fundamentally changed how HCV infection is managed in clinical settings worldwide. Additionally, these studies provided a detailed understanding of the on-going evolution of the HCV genome, and the role that host factors play in influencing the outcome of infection (Table 2).

**Table 2 pathogens-11-00306-t002:** Major findings from the Irish anti-D cohort.

Study Type	Major Research Findings
Clinical and Molecular Epidemiology	The HCV sequence in Irish anti-D Ig recipients arose from a single strain, distinct from circulating HCV strains in Ireland, confirming single-source outbreaks due to the use of contaminated blood in the preparation of anti-D Ig [11,18].
	The development of liver pathology and HCV disease progression is slow in the absence of additional lifestyle risk factors [8,19].
	There is a low risk of transmission to children born around the time of their mother’s infection, but transmission to other family contacts is less likely [20].
Viral Genetics	Immune pressure is the dominant driver of viral evolution early in infection, while evolution towards a consensus ancestor sequence (with higher fitness) is dominant in the late stages of chronic infection [21].
	The major HCV genotypes were estimated to have diverged between 500–2000 years ago, requiring the on-going transmission of HCV in historical human populations [22].
	Evasion of host immune pressure comes at a significant cost to viral fitness, and following transmission events the virus reverts to a consensus sequence with higher fitness [23].
Infection and Immunity	The HLA-DRB1*01, -DQB1*0501, -DRB1*0401, HLA-DRB1*15, -A*03, -B*27, and -Cw*01 alleles are associated with spontaneous resolution of HCV infection [24,25].
	Immune pressure requires both a protective host HLA allele and a specific viral immunodominant epitope—the HLA-B*27 allele is only protective in genotype 1 HCV infection and is neutral when the infectious strain harbours escape variants [26,27,28].
	Polymorphisms in innate immune genes (IFNL3, KIR2DS3, and IFIH1) are associated with the spontaneous resolution of HCV infection [29,30].

## 4. Clinical and Molecular Epidemiology Studies

Initial studies performed on individuals who received suspected contaminated batches in 1977 confirmed the presence of a single viral genotype (1b) in this population, distinct from the mixture of genotypes circulating in other at-risk populations in Ireland at the time [20]. Viral sequences (from the 5′ untranslated region and NS5 region) from these individuals and a contaminated anti-D immunoglobulin batch clustered on phylogenetic analysis, implicating this event as a single-source outbreak [20]. A similar phylogenetic study confirmed the presence of a single viral genotype (3a) in women exposed to anti-D Ig manufactured between 1991 and 1994 [11].

Early screening efforts rapidly identified hundreds of recipients of the contaminated anti-D Ig, who had subsequently developed chronic HCV infection (antibody positive and PCR positive using diagnostic assays), as well as substantial numbers of recipients who spontaneously cleared the infection (antibody positive and PCR negative using diagnostic assays). Subsequent histological studies and epidemiological follow-up confirmed that women who were antibody positive but PCR negative had spontaneously cleared the virus and showed no evidence of progressive liver disease [31,32]. Amongst the individuals who did become chronically infected, liver biopsies taken between 17–18 years post-infection showed remarkably little evidence of liver pathology [19,33,34]. More than 85% of individuals had overall Ishak scores (measures of fibrosis) <5 and at a histological level, these women were largely indistinguishable from patients in whom liver histology would be expected to be normal [19,33]. These studies represented some of the first concrete epidemiological data on disease progression and confirmed anecdotal evidence that liver pathology can be a very slow and gradual process in the absence of additional risk factors, particularly in women [15,35].

Due to the infection of pregnant women, there was considerable concern over possible vertical transmission of HCV. Investigations found that 13 of 1338 children born to HCV+ mothers had serological evidence of infection (0.97%). Three of these children were found to be PCR positive and infected with genotype 1b HCV [18]. This represents a risk of transmission to children born around the time of their mother’s infection of 0.35% [18]. No evidence for sexual transmission or family-contact transmission was found [18]. These findings closely correlate with similar studies of the German anti-D HCV outbreak [9] and highlight the low risk of HCV transmission to family contacts in the absence of direct blood contact.

## 5. Viral Genome Evolution Studies

Estimating how long a virus has been spreading in historical human populations deepens our understanding of the interactions between the virus and the human host, and how the virus is transmitted through generations. HCV is considered a recent human pathogen, however it has been unclear whether this virus was maintained within specific endemic regions prior to its spread due to modern transmission routes. As the source viral sequence from the Irish anti-D Ig HCV outbreak was known, viral sequences from infected individuals taken decades after the original exposure could be used to estimate the rate of viral sequence change, and infer the evolutionary history of HCV [22]. The rates of synonymous substitution for the NS5 and E1 genes were estimated at 1.1 × 10^−3^ and 1.7 × 10^−3^ per site per year, respectively, while the nonsynonymous rates were lower for both E1 (4 × 10^−4^) and NS5B (6 × 10^−6^) [22]. From these rates, the divergence of HCV subtypes 1a and 1b was estimated to be about 300 or more years ago, with the major genotypes diverging at least 500–2000 years ago [22].

The rate of evolution of the viral genome differs across different genome regions [36]. The hypervariable region (HVR) within the HCV E2 region, which encodes a viral glycoprotein, is of particular interest as this viral protein is immunodominant and antibodies targeting this region can effectively neutralise the infectivity of HCV viral particles. The ability of HCV to escape immune pressure through variations in the HVR is important for the establishment of persistent infection during the acute phase following exposure. Within the anti-D Ig cohort, all individuals had a single major HVR variant, however no two individuals had the same HVR sequence representing divergent evolution from the original viral inoculum [23,37]. These sequence changes in the HVR were not random, indicating that the HVR evolves via immune pressure in a framework of constraints imposed by the viral genome structure [37]. Of interest, the rate of HVR evolution is higher within an individual compared with between individuals, indicating that host-specific reversion events preferentially occur in this region following viral transmission to a new host [36].

Immune pressure represents a major driver of patient-specific viral evolution with immune epitope amino acid substitutions directed away from the consensus sequence in individuals with epitope-specific human leukocyte antigen (HLA) alleles (immune selection), and towards the consensus sequence in individuals lacking epitope-specific HLA alleles (reversion) [23]. The observation that reversion, in the absence of selective immune pressure, is a frequent event highlights the significant fitness cost of immune escape and implies that vaccination using consensus sequences may form the highest barrier to viral escape (and subsequently the best protection) [23]. Viral evolution also varies depending on the stage of disease. By examining viral sequences at three distinct time-points, Bailey and colleagues identified that cytotoxic T lymphocyte (CTL) immune pressure was a dominant driver of viral evolution early in infection, while evolution towards a consensus common ancestor sequence was evident in the late stages of chronic infection [21]. This study concluded that ongoing HCV evolution late in infection is not random genetic drift; instead, it represents selective pressure towards a common ancestor and evolution away from the inoculum sequence [21]. This viral evolution late in chronic infection balances replicative fitness and immune escape. The cohort has also proved valuable in determining how the host immune response shapes HCV viral genome evolution. In particular, focus has been on the IFNL4 locus [38].

## 6. Infection and Immunity Studies

The impact of host genetics on viral infection and the host immune response is difficult to assess. Clinical heterogeneity (such as ethnic backgrounds, age at infection, viral genotype, and mode of infection) hampers the direct comparisons between human cohorts. In this sense, the Irish anti-D Ig HCV outbreak provides a unique cohort of homogeneous individuals and allows for the investigation of host immune factors without several confounding factors that influence immune responses.

Several immunodominant immune epitopes were identified in the HCV genome, yet despite the ability of the immune system to recognise HCV, many individuals are unable to clear HCV infection. HCV employs several strategies to subvert the host immune response and, in doing so, can persist within an infected liver. Chronic HCV infection induces immune dysfunction in a range of immune cell subtypes, including dendritic cells (DC) [39], natural killer (NK) cells [40], B cells [41], and T cells [42]. Human monocytes recognise the HCV NS4 protein and express two potent immune regulatory cytokines: interleukin-10 (IL-10) and transforming growth factor-β (TGF-β) [43]. This monocyte-derived IL-10 and TGF-β suppress the production of the pro-inflammatory cytokines interferon-γ (IFNγ) and IL-17 by Th1 and Th17 HCV-specific T cells [43]. Chronic HCV infection also results in dysfunctional immune signalling via the degradation of signal transducer and activator of transcription (STAT)1 and STAT3 proteins, which reduced signalling following type I IFN stimulation, and may be why some individuals fail to respond to IFN-based therapies [44].

While immune dysfunction is evident in chronic infection, this fails to address the question of whether differences in immune responsiveness influence the outcome of acute infection (i.e., resolution or persistent infection). The Irish anti-D Ig HCV outbreak provides the opportunity to perform studies investigating genetic associations between immune-related genes and spontaneous resolution of HCV infection. The first study of this type focused on the HLA complex, which encodes a highly variable array of important antigen-presenting molecules that are known to influence host response to infection. The HLA-DRB1*01 allele was associated with individuals who spontaneously cleared the virus, compared to those who developed chronic infection (27.4% vs. 7.1%, *p* = 0.001, odds ratio OR = 4.9) [24]. Subsequent studies also identified associations between the HLA-DRB1*0401, HLA-DRB1*15, HLA-DQB1*0501, HLA-A*03, HLA-B*27, and HLA-Cw*01 alleles and viral clearance [25,45]. Of all the alleles, HLA-B*27 had the strongest association (odds ratio (OR) 7.99) [45].

This HLA-B*27 allele allows the presentation of a single immunodominant viral peptide epitope (spanning residues 421–429 of the NS5B protein) to CD8+ T cells, yet despite the strong immune pressure, escape variants are rarely seen [26]. Artificial variants that disrupted HLA-B*27 binding could completely abolish T-cell cross-recognition; however, these viral variants showed a dramatic reduction in replication efficiency [26]. The protective HLA-B*27 allele is dependent on the viral genotype; non-genotype 1 HCV strains lack the consensus immunodominant HLA-B*27 epitope and the protective influence of the HLA-B*27 allele is lost [27]. Certain genotype 1b HCV strains also harbour immune escape variants in this epitope, and the HLA-B*27 allele has no protective influence during infection with these viral strains [46]. By comparing multiple individuals with the same HLA alleles, it is possible to identify a ‘molecular footprint’ of immune pressure and identify novel epitopes [47,48]. This information is particularly important for the development of HCV vaccination strategies to generate immune responses specific for immunodominant epitopes that will be recognised by most of the vaccinated population.

While the HLA complex is a major determinant of susceptibility, other polymorphisms in immune-related genes, which correlate with the outcome of HCV infection, were identified from the Irish anti-D Ig HCV outbreak. Dring and colleagues investigated the impact of NK cell-associated killer cell Ig-like receptor (KIR) genes, as well as a single nucleotide polymorphism (SNP) near the type III IFN gene, IFNλ3 (previously IL28B; rs12979860) [29]. They found that the NK cell gene KIR2DS3 and the IL28B ‘T’ allele were associated with chronic infection (odds ratio (OR) 1.90 and 7.38, respectively) [29]. Hoffmann and colleagues investigated polymorphisms in IFIH1 (interferon induced with helicase C domain 1; also known as MDA-5), an intracellular receptor that recognises viral entry into a host cell [30]. The IFIH1 signalling pathway is critical for the recognition of HCV infection and is suppressed by the virus. Two polymorphisms in the IFIH1 gene correlated with the spontaneous resolution of infection, due to an increased antiviral response [30].

While substantial information has been garnered from studying these women on SRs and CIs, the ESNs in the Irish anti-D cohort have yet to be studied.

## 7. Viral Resistance in Other Cohorts: What Has Been Learned and What Are the Opportunities?

Most work on ESN cohorts to date has been carried out on people who inject intravenous drugs (PWIDs), who are known to share needles and other injection paraphernalia with those who are HCV positive (Table 3) [49]. Typically, this high risk behaviour results in a low dose exposure to HCV infection that results in chronic infection [49]. Despite this, these ESN individuals fail to contract the infection themselves. Studies, to date, of ESNs in PWID cohorts have found interesting differences in key cellular and secreted processes [49,50,51].

Most work in PWID ESNs centred around NK cell functionality. Lloyd and co. reported increased counts of CD69 + CD56dim and CD69 + CD56bright NK cells—indicating a more ‘active’ phenotype [50]. They also showed greater IFNγ production and less CD107a (a degranulation marker) upregulation in NK cells derived from ESN donors in vitro [52]. Other NK cell work carried out on Canadian PWIDs found no difference in NK cells at baseline [53]. Nor did they find differences in IFNγ production by peripheral blood mononuclear cells (PBMCs), as measured by ELISpot in response to HCV peptides^53^. This finding has been contradicted in other studies on ESN cohorts, where robust antigen-specific T-cell responses as measured by ELISpots have been found, implicating enhanced T-cell responsiveness as the mechanism by which individuals are protected from HCV infection [52,54,55,56]. Additionally, reported in PWID ESN cohorts is an increased frequency of KIR2DL3 + NKG2A-NK cell populations compared with healthy controls, individuals who were previously chronically infected and cleared the infection with therapy (sustained virological responders; SVR) and SR populations [57]. This cell population is uninhibited by HLA-E ligation and so produces more IFNγ. KIR2DL3 HLA-C1 homozygosity has also been observed [58].

Fewer studies on the cytokine profiles of these individuals were carried out, although there are reports of increased IL-6, IL-8 and tumour necrosis factor α (TNFα) in the serum [49]. SNPs in the IL-12B gene that increase IL-12 production were also associated with resistance [59]. NK cells from exposed seronegative cohorts produce more IFNγ and TNFα on stimulation compared with either SR or SVR groups [60].

Similar to the CCR5 mutation seen in HIV-1 resistance, it would be plausible to think a similar observation might be made in a human population that resists HCV infection, and perhaps a polymorphism in CD81 the receptor for HCV might explain the resistance [61]. However, this does not appear to be the case with HCV. Studies by two groups found CD81 to be very highly conserved with no genetic alterations [62]. Other work found SNPs in both claudin-6 and occludin in two ESN individuals, though later functional studies showed that these SNPs did not confer resistance in vitro [63,64]. Further work refuting the idea that HCV resistance is mediated by entry receptor mutations comes from Matthew Cramp’s group at Plymouth. Through a look back study into 1340 individuals who received a HCV-contaminated blood transfusion, Cramp’s group identified and recruited 8 individuals with an ESN phenotype. Whole exome sequencing of this group showed no enrichment for SNPs in any HCV entry receptors [65].

Further work on this cohort corroborates studies on PWIDs and shows enhanced NK cell functionality with increased NKp30, NKp80, and KIR2DL3 expression and increased cytotoxic responses with IL-2 [49]. Occupational exposure to HCV via needle stick injury demonstrated early NK cell activation and increased NKp44, NKp46, and CD122 expression as important mediators of resistance to infection [66]. Pilot work by our group on a smaller cohort exposed to both genotype 3a and genotype 1b of HCV via contaminated anti-D found that ESNs had decreased monocyte counts as well as increases in IL-8 and IL-18 compared with SR and SVR [67]. We did not observe an increase in NKp30 expression [67].

**Table 3 pathogens-11-00306-t003:** Summary of results from previous ESN studies.

Route of Exposure	Mechanisms of Resistance
Blood Transfusion	NK cell counts were reported to be increased, alongside an increase in NK cell functionality and increased NKp30, NKp80, and KIR2DL3 expression on both NK cell subsets. In response to IL-2, NK cells from ESNs have increased levels of cytotoxicity. Some reports of detectable IFNγ ELISpots in response to HCV peptides [49].
	No enrichment for SNPs in HCV entry receptors in ESNs [65].
Occupational Exposure	ESNs following needle stick injury had early NKT activation and increased serum cytokine responses. ESNs also had increased CD122, NKp44, NKp46, and NKG2A expression, cytotoxicity, and IFNγ production. This robust response correlated with a strong HCV-specific T cell response [66].
	ESNs following needle stick injury in Germany had robust CD4+ T-cell response to HCV [68].
	Robust anti-HCV T-cell response also noted in needle stick injury ESN individuals as measured by ELISpot for IFNγ [66].
People Who Inject Drugs	ESNs had distinct lipidomic profiles compared to HCV-susceptible individuals [69].
	ESNs had increased counts of CD69 + CD56dim NK cells and an increased number of NKp30+ CD56bleft CD16+ NKs with greater IFNγ production, but less CD107a expression. ESNs had no difference in their ELISpot responses compared to CIs. There was no association between IL28B, HLA-C, or KIR2DL3 and resistance to HCV infection [50,70].
	PWID ESNs had robust HCV-specific T-cell responses as measured by ELISpot [52,54,56].
	ESNs had increased KIR2DL3 + NKG2A- NK cells compared to controls, CIs and SRs. These NKs are not inhibited by HLA-E ligation and therefore produce greater IFNγ in response to stimulation [57].
	The 1188A/C polymorphism of IL-12B, C allele, and CC genotype are associated with HCV resistance [58].
	Claudin-6 and occludin variants were found in an ESN individual but were not sufficient for resistance in vitro. CD81 appears to be very highly conserved with no genetic alterations found in a study of ESNs cases [62,63,64].
	The IL28B genotype rs12979860 CC is not associated with resistance to HCV, but ESNs have higher homozygosity for KIR2DL3 HLA-C-1 [53,58].
	Increased IL-6, IL-8, and TNFα in the ESNs compared with controls, Cis, and SRs [49].
	Enhanced IFNγ, TNFα production, and degranulation by CD56dim NK cell subsets from ESNs [60]. No baseline differences in NK cells from ESN, SRs, and HCV CIs in Canadian IVDUs [53].
Anti-D Cohorts	Decreased monocyte counts in ESNs. Increased IL-8 and IL-18 in ESNs. Enhanced NK cell function—greater IFNγ production [67].
	No significant increase in NKp30, or changes in CD56bleft or CD56dim NK cell counts. No difference in degranulation, but CD56dim NK cells produced more IFNγ when stimulated [67].

Given that the Irish anti-D cohort is comprised of women who were otherwise healthy at the time of infection, we propose that they are an ideal study group in which to interrogate resistance to viral infection.

## 8. Conclusions

It is virtually impossible to study natural resistance to viral infection in humans as controlled infection studies are usually unethical, and it is difficult to be sure that uninfected individuals in the general population were ever exposed to the virus. Iatrogenic infections, such as the contaminated anti-D episode in Ireland, when approximately 600 women did not become infected despite being exposed to anti-D with high viral loads of HCV, provide a valuable opportunity to probe the mechanisms of resistance.

The nature of this outbreak, wherein all HCV-exposed individuals were pregnant females, potentially limits the extrapolation of findings from the cohort to other less homogenous groups. Several sex differences in the immune systems of males and females were described, including in Toll-like receptor (TLR) expression, the type I IFN system, and the ability to spontaneously resolve HCV infection [71,72,73,74]. Given the Irish anti-D cohort are all female, phenotypes described using data from these donors may not apply directly to HCV-exposed males.

Additionally, all donors in this cohort were exposed to a single source of genotype 1b HCV. HCV genotypes exhibit different infectivities and responses to therapy [75]. While exposure to a single HCV genotype has been a strength of the anti-D cohort, the applicability of findings to other HCV genotypes is limited and warrants continued investigation.

Understanding the immune mechanisms responsible for viral resistance is vital for future therapeutic and vaccination strategies, both against HCV and unrelated human viral infections. The high cost of anti-viral therapies targeting HCV and the ease of HCV transmission within at-risk populations means that a vaccination strategy is required for HCV eradication [76]. Studies of ESN cohorts inform us about alternatives to the robust long-lived T-cell responses in providing resistance to HCV infection and highlight the role of innate immunity in human viral resistance. A greater understanding of innate immune mechanisms enables the development of novel therapeutics capable of boosting innate immunity to combat acute viral infection and aid in the development of viable HCV vaccines.

Since 2010, curative, direct acting antiviral (DAA)drugs for HCV have been available and deployed with great success in developed countries, particularly in the chronically infected members of the Irish cohort. Nowadays, this provides a useful opportunity to study the impact of long-term infection on the immune system post curative therapy. The risk of HCC following DAA treatment and SVR is incompletely eliminated and emerging work suggests an ongoing disruption to the immune response, even in the absence of antigenic stimulation [77]. Work on liver tissue from DAA-treated SVRs and healthy control donors showed an altered transcriptomic and epigenetic landscape that persisted following HCV clearance. The pharmacological modulation of the epigenetic landscape shows promise in reducing HCC risk following SVR with DAAs [78]. Khera et al. showed that, despite early therapeutic intervention and rapid viral clearance of persistent disruption levels, several soluble inflammatory mediators could be observed. Even a short duration of viraemia can cause persistent immune changes with yet to be determined consequences [79]. Several cell populations, particularly mucosal associated invariant T (MAIT) cells, fail to fully recover to their pre-HCV levels and function. NK cell diversity has also been reported to be altered [80]. The consequences of these persistent immune changes are yet unknown, but may impact on the induced immune response to infection and vaccination and so warrant further study. Well-described and relatively controlled cohorts, such as the Irish anti-D cohort, could be useful in addressing some of these outstanding questions about resistance to infection and the restoration of immune homeostasis following the clearance of HCV infection.

## Figures and Tables

**Figure 1 pathogens-11-00306-f001:**
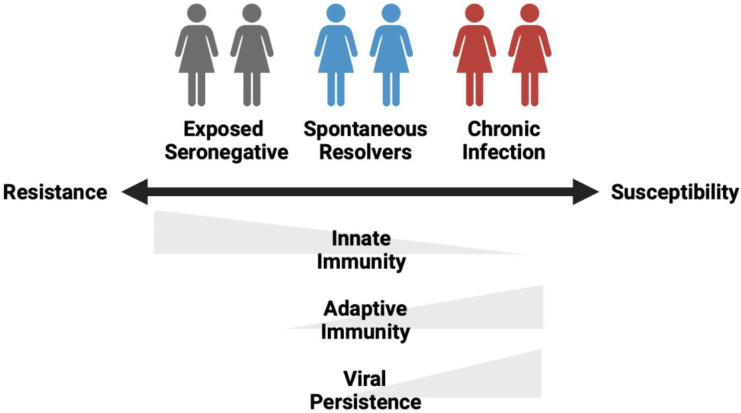
Spectrum of outcomes following exposure to HCV.

**Table 1 pathogens-11-00306-t001:** Details of the HCV 1977–1979 outbreak from the Irish Blood Transfusion Service (IBTS). (Anon. 2012. National Hepatitis C Database for Infection Acquired through Blood and Blood Products: 2012 Report. Retrieved (http://www.hpsc.ie/AZ/Hepatitis/HepatitisC/HepatitisCDatabase/BaselineandFollow-upReports/; accessed on 1 December 2021). A total of 12 contaminated batches were identified from the 1977–1979 outbreak. A total of 6 of the infected batches that were associated with seropositivity of 30% or greater were classified as high risk.

Details	1977–1979 Outbreak
No. of contaminated batches	12 (4062 vials)
No. of high-risk batches (>30% Ab+)	6
Genotype of HCV involved	1b
No. of chronically infected women from HR batches	356
No. of spontaneous resolvers from HR batches	326
No. of potentially resistant (ESN) women from HR batches	611

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
