# Peer review of "Uncovering Resistance to Hepatitis C Virus Infection: Scientific Contributions and Unanswered Questions in the Irish Anti-D Cohort"

_pathogens, 2022, doi:10.3390/pathogens11030306_

Round 1

Reviewer 1 Report

This is a valuable review that summarizes research based on a cohort of healthy Irish women who were inadvertently exposed to HCV infection in the 1970s, and benefits from the long time-course of the studies to shed light on virus evolution, host defense and the development of liver disease. The low frequency of disease progression, and in particular of HCC in these healthy subjects provides much reassurance on the long-term outcome of untreated infection, and might raise questions about whether all HCV exposure events need to be followed by anti-viral drug treatment. The authors have kept away from this potentially controversial issue, but it would be good to know their perspective.

The conclusions concerning virus evolution within the infected host, the fitness cost of immunoselection and the reversion of virus sequences towards consensus in chronic infection are arresting and deserve to be more widely appreciated.

The review describes data showing that HCV can be immunoselected by host T cell immunity, but acknowledges that the virus also directly inhibits T cells as well as antigen-presenting cells. The question arises whether HCV strategies to disable T cells are really important, given that the virus can evade antigen-specific T cells by mutation. Do the authors think that the reversion of viruses towards consensus sequences in later infection can occur only as a result of T cell subversion?

This is an important contribution as-is, but would be enhanced if authors were to use the data to raise more challenging questions, to which answers may not yet be clear.

Author Response

Many thanks for reviewing the manuscript. 

Regarding your question about whether all HCV individuals should be treated based on the low progression to HCC seen in the Irish cohort- I would be cautious about extrapolating findings on the low progression rate to HCC in this cohort to others- this women were all healthy at the time of infection, something which is not a feature of other HCV cohorts (Intravenous drug users and so on). A recent 35 year follow up study on the cohort noted that progression to liver disease increases dramatically at the 20-25 year mark. It isn't yet possible to screen donors based on likelihood to develop liver disease, therefore it is prudent to treat all individuals diagnosed with HCV. 

We also found the studies investigating host immune pressure and HCV immune escape to be interesting and worth highlighting. 

Regarding the importance of T cell suppression mechanism in conjunction with escape mutations - it seems that many viruses have a multi pronged approach to subvert and evade the immune system. It provides a fail safe should HCV encounter a host that it cannot evade appropriately. 

Thank you for your final comment. We have updated the manuscript based on your comments and others and feel it has improved. 

Reviewer 2 Report

Comments to the authors:

The review article by Sugrue et al., presents an account of the HCV outbreak that took place between 1977-79, in which HCV-contaminated anti-D ig preparations were administered to approximately 1,700 pregnant Irish women. Moreover, the authors provide a summary of the studies that have been performed employing this cohort with the aim to characterize immune responses against HCV.

The combination of historical account and translational science makes the review article quite interesting to read. The reviewer considers that indeed this particular cohort represents a useful resource for the characterization of HCV infection and its associated complications. However, many aspects of the article feel rushed. As detailed in my specific comments, I consider that in particular the tables and the conclusions section need to be improved.

In the future, please consider adding line numbers to the manuscript in order to facilitate communication between authors and reviewers.

Major comments:

  • There are two tables titled “Table 2”.
  • Table 2 (second one) needs to be improved, as in its current state reads like randomly assembled facts and not as digested information that is presented in an structured fashion. Moreover, in one instance there is just the title of an article pasted into the table! (i.e., “Claudin-6 and Occludin Natural Variants Found in a Patient Highly Exposed but Not Infected with Hepatitis C Virus (HCV) Do Not Confer HCV Resistance In Vitro”).
  • The reviewers agrees with the authors when they state that the Irish anti-D cohort represents a very interesting group of patients to study. However, it would be useful to include in the conclusions sections the limitations that this cohort presents. For example, all patients were of the same sex, a factor that has been described to influence immune responses (Micallef et al., J. Viral Hepat. 2005 PMID 16364080, Grebely et al., Hepatology 2014 PMID: 23908124).
  • In the conclusion section, the authors discuss the impact of long-term infection on the immune system post curative therapy. The authors could cite Hamdane et al., Gastroenterology 2019 (PMID 30836093) and Jühling et al., Gut 2021 (PMID: 32217639) as examples of the type of analysis that could be carried on distinct immune populations.

Minor comments:

  • Replace ethic by ethnic (page 2, “predominantly of an Irish ethic origin”).
  • Add meaning of ESN, EU, HC and CI.
  • The meaning of this paragraph on page 3 is not quite clear. Please consider rewriting and dividing it into shorter sentences. “These studies have contributed to an epidemiological evidence-base that has fundamentally changed how HCV infection is managed in clinical settings worldwide as well as providing a detailed under-standing of the on-going evolution of the HCV genome and the role that host factors play in influencing the outcome of infection (Table 2)”.
  • Reformat this reference in table 2: “Barrett, Ryan, and Crowe 1999; McKiernan et al. 2000, 2004”.
  • Please consider modifying the following sentence, so it reads: “Immune pressure requires both a protective host HLA allele and a specific viral immunodominant epitope – the HLA-B*27 allele is only protective in genotype 1 HCV infection and is neutral when the infectious strain harbours escape variants”.
  • Add “as” to the following sentence on page 4, so it reads: “developed chronic HCV infection (antibody positive and PCR positive using diagnostic assays) as well as substantial numbers of recipients who spontaneously cleared the infection”.
  • On page 5, move a comma, so the sentence reads: “From these rates, the divergence of HCV subtypes 1a and 1b was estimated to be about 300 or more years ago”.
  • On page 5, add “that”, so the sentence reads: “which encodes a viral glycoprotein, is of particular interest due to the fact that this viral protein is immunodominant and antibodies targeting this region can effectively neutralise the infectivity of HCV viral particles”.
  • On page 5, add meaning of HLA and delete it from page 6.
  • On page 6, add meaning of CTL.
  • On page 6, add meaning of IL-10, TGF-β and IFNγ.
  • Add meaning of NK on page 6 and delete it on page 7.
  • Add meaning of SNPs on page 7.
  • On page 7, consider modifying the following sentence: “Nor did they find differences in the expression of IFNγ production from PBMCs as measured by ELISpot in response to HCV peptides”. Probably “expression of” could be deleted.
  • On page 7, add meaning of SVR, SR and TNFα.
  • In table 2, write “difference” instead of “diff”.
  • Dot missing at the end of this sentence: “Decreased monocyte counts in EU. Increased IL-8 and IL-18 in EU. Enhanced NK cell function - greater IFNγ production 67”.
  • On page 9, delete a comma, delete “virus” and add “a” so that the following sentence reads: “when many women did not become infected despite being exposed to anti-D with high loads of HCV, provide a valuable opportunity to probe the mechanisms of resistance”.
  • In the conclusions section, reformat these references:

https://academic.oup.com/jid/ad-vance-article/doi/10.1093/infdis/jiab048/6124746

https://www.nature.com/articles/s41467-018-04685-9

  • Add the meaning of MAIT in the conclusions section.
  • Dot missing at the end of the conclusions section.

Author Response

Thank you for reviewing the manuscript. 

We have updated the table, in particular table 3 to be more articulate and less rushed. 

Thank you for the suggests regarding the conclusions section, we have updated the section to include the limitations associated with the Irish anti-D cohort (sex, genotype and lifestyle). We have also included the suggested references on epigenetic changes persisting in SVRs. 

Thank you too for your minor comments. We have updated the manuscript to correct these typos and address your comments. 

Reviewer 3 Report

This review article decribes the history of the cohort infused with the HCV-contaminated anti-D blood between 1977-1979 and the research outcomes derived from studying this cohort, which has deepened the understanding of HCV. It is well organized and clearly written.

Author Response

Many thanks for your positive feedback on our manuscript. 

Reviewer 4 Report

From the abundance of data presented in this review by Jamie A Sugrue and Cliona O'Farrelly, it is sometimes difficult to recognize and not clear what exactly the role of the anti D cohort was for our current understanding of HCV infection (scientific contribution), or which observations are exactly based/related to the anti D cohort. The review would benefit from a clearer presentation of the specific findings from the anti D cohort and a clearer discussion of the results from the anti D cohort in the context of observations from other studies/cohorts (consistencies and discrepancies). Tabular presentation(s) of the data discussed are needed which summarize the complex observations better. Also the potential limitations of studies from this cohort should be discussed, gender, genotype, etc.

Figure 1: add percentages to indicate proportions of patients per outcome (based on anti D cohort)

No. of high risk batches (>30% Ab+) – need to be explained in the table legend

Page 3, viral genetics: „Evasion of host immune pressure comes at a significant cost to viral fitness and following transmission events the virus reverts to a consensus sequence with higher fitness“ Is this statement really primarily due to the anti-D cohort and the paper mentioned? I think the paper cited is not directly about viral fitness but rather evolutionary dynamics. The statements and papers cited in the table should be specific to the anti D cohort.

„A greater understanding of innate immune mechanisms will enable the development of novel therapeutics capable of boosting innate immunity to combat acute viral infection or maximise vaccine efficacy.“ Sounds like a vaccine is available here? Should be worded differently.

Add table on patient demographics from the anti D cohort highlighting specific features potentially related to observations discussed in the manuscript, e.g. slow disease progression

Add table on virus and host factors of the anti D cohort related to virus evolution and immune responses (as discussed in the manuscript)

Minor points:

Page 6: is able to persistent within an infected liver

Page 7: as well as the a SNP near the type III IFN gene IFNλ3

Add a list of abbreviations and check the text again for the correct introduction of abbreviations

Author Response

Thank you for your comments. 

We have updated the tables in the manuscript. We have added in the cohort limitations to the discussion (including sex, genotype and lifestyle). We have added a sentence to indicate that the overall health of the cohort likely contributed to the slow disease progression described. 

We have added a legend to explain the low and high risk classification system for the infected batches. 

We have updated the reference in the viral genetic section to be one from the anti-D cohort. We have reworded the vaccine sentence to make it clear that no approved vaccine yet exists. 

We have updated the abbreviations in the text. 

Round 2

Reviewer 2 Report

The review article by Sugrue et al., has been considerably improved with the modifications provided by the authors. The reviewer has only very minor additional suggestions for the authors:

Minor comments:

  • In the tables, please add a space between the last word of the sentence and the reference numbers, so they have the same format as the rest of the article.
  • Add meaning of SNP on page 6 line 234 and delete it on page 7 line 270.
  • In the first box of table 3, correct the first sentence so it reads “have been reported”. Also replace IL2 by IL-2 and IFNg by IFNγ.
  • Some of the sentences in the tables end with a point and other don’t.
  • Add meaning of TLR and replace interferon by IFN on page 9 line 308.
  • Add abbreviation of DAA on page 10 line 326.

Author Response

Many thanks for the minor edits, I have updated the manuscript with your edits. 

Reviewer 4 Report

No additional comments.

Author Response

Many thanks for the final comment.